# Status of Sustainability Development of Deep-Sea Mining Activities

Wenbin Ma [1], Kairui Zhang [1], Yanlian Du [1], Xiangwei Liu [2] and Yijun Shen [1],*

1   State Key Laboratory of Marine Resources Utilization in South China Sea, Hainan University, Haikou 570228, China
2   Logistic Engineering College, Shanghai Maritime University, Shanghai 201306, China
*   Correspondence: 995390@hainanu.edu.cn

**Abstract:** With technological improvement such as ore exploration, robotics, and hydrodynamic lifting, deep-sea mining has attracted more attention from governments, companies, and scientific research institutions. Although its research and development has made great progress, there are still many obstacles in its industrial development, such as environmental pollution and sustainability development issues. This article analyses the research status of the sustainable development of deep-sea mining from an overall perspective. Through a literature review, this paper also discusses the application of the full life cycle assessment method to analyze environmental impact during the entire process of deep-sea mining ore application. Overall, this paper summarizes the research gaps that exist in the sustainable development of deep-sea mining, including the lack of sufficient quantitative research, environmental baseline data research, cumulative environmental impact assessment, resource recycling technology, and acceptable environmental impact range analysis. The significance of this article is to point out the most urgent problems to be solved in the research direction of the sustainable development of deep-sea mining in current academic circles. It has far-reaching potential to promote the industrialization process of the entire deep-sea mining industry.

**Keywords:** deep-sea mining; sustainability; life cycle assessment; environmental baseline data; environmental–social pressure

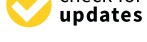



## 1. Introduction

Deep-sea mining can be defined as: the utilization of hydrodynamic or mechanical methods to transport mineral ores from the seabed to the ocean surface and then transport ores to the land-based processing plants by ships [1]. Although the concept of deep-sea mining was proposed in the 1960s, its industrialization has still not been realized [2,3]. The environmental pollution and sustainable development issues of deep-sea mining have become the biggest restraints on its improvement [4,5]. Figure 1 describes a schematic diagram of one typical deep-sea mining project [6,7]. The structures involved consist of seabed mining vehicles, a vertical lifting system, a production support vessel, bulk carriers, a mineral ore processing and refining plant, etc., and Figure 1 also describes the various environmental impacts to the seabed, water columns, and ocean surface.

Giurco and Cooper [8] utilized the mineral resources landscape approach to analyze the sustainable development of mining activity from both global and local perspectives. Considering social, ecological, technological, economic and governance aspects, the research identified that the role of mineral recycling technology is overlooked in deep-sea mining. Roche and Bice [9] qualitatively analyzed the key technologies for the sustainable development of seabed mining, gave more attention to the social impact. The social impacts considered in this paper include social economy, human rights, the history and experience of terrestrial mining activities, and the issues related to land use, ownership, local traditional fishery industries, etc. The study reveals the importance of a valid and

comprehensive set of assessment indicators for the sustainable development of deep-sea mining. Carvalho [10] conducted research on the sustainable development of terrestrial mining. The research concludes that most terrestrial mining activities need to be improved to meet the requirements of the local community, as well as environmental health and sustainable development objectives. The research emphasizes the function of environmental impact assessment (EIA), which could be used to estimate the gains, costs, losses and consequences of mining activities. Glover et al. [11] analyzed deep-sea mining sustainability from the view of '*blue economy*', which considers the regulatory oversight setting targets of taxonomic data delivery. The research utilizes a taxon-focused approach, which is different from the traditional ecosystem-based management approach. Levin et al. [12] carried out a systematic literature review on deep-sea mining sustainability challenges. The research considers the environmental, legal and social aspects of sustainability evaluation. It shows that the sustainable development of deep-sea mining has many research gaps, and advocates slowing down the industrial exploitation of seabed minerals in the very near future.

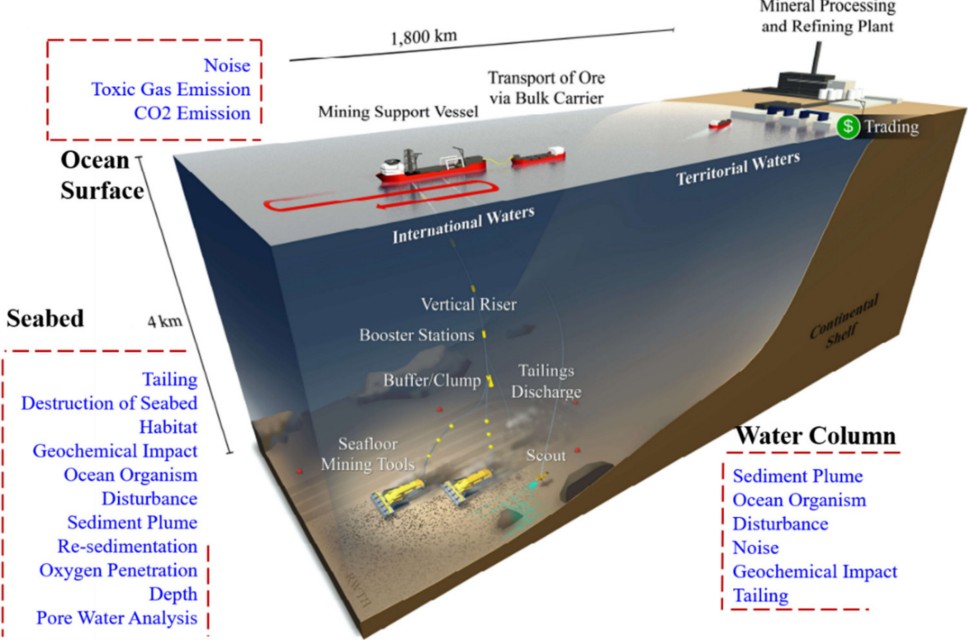

**Figure 1.** Schematic diagram of a typical deep-sea mining project [6,7].

Through a systematic literature review, we show that the sustainable development of deep-sea mining is a relatively new and immature topic. Most of work completed only involves qualitative analysis and discussion. Most of the theories come from similar industries such as terrestrial mining analysis and offshore oil and gas development. The objective of this paper is to analyze the research status of the sustainable development of deep-sea mining (mainly focusing on polymetallic nodule mining activities). It attempts to discuss the application of the life cycle assessment method to analyze the environmental impacts during the entire process of deep-sea mining ore application, and summarizes the research gaps in this field. The paper is arranged as follows: Section 2 explains the sustainability definition and motivation of deep-sea mining activities. In Section 3, the life cycle assessment approach is applied to a deep-sea mining project to analyze its environmental–social impact. Section 4 discusses the existing issues and challenges in the sustainable development of deep-sea mining. Then, in Section 5, conclusions and recommendations are given.

## 2. Deep-Sea Mining Sustainability

### 2.1. Motivation of Deep-Sea Mining

Deep-sea mining is a mineral retrieval process from the seabed to the ocean surface; then, the collected minerals are transported to land-based ore processing plants utilizing seabed mining vehicles, vertical lifting equipment, production support vessels and bulk vessels [5]. The major reason to propel deep-sea mining industrialization is the contradiction between the increasing demand for high value-added ores for rapid economic development and the shortage of terrestrial mineral reservation [13]. The deep ocean seabed has enough ore reserves for the world to consume for hundreds, even thousands, of years. This is the major motivation for deep-sea mining. The typical minerals for deep-sea mining consist of polymetallic manganese nodules (most popularly focused), polymetallic sulphides, and cobalt-rich ferromanganese crusts, located in different sea areas (see Figure 2) [14–17].

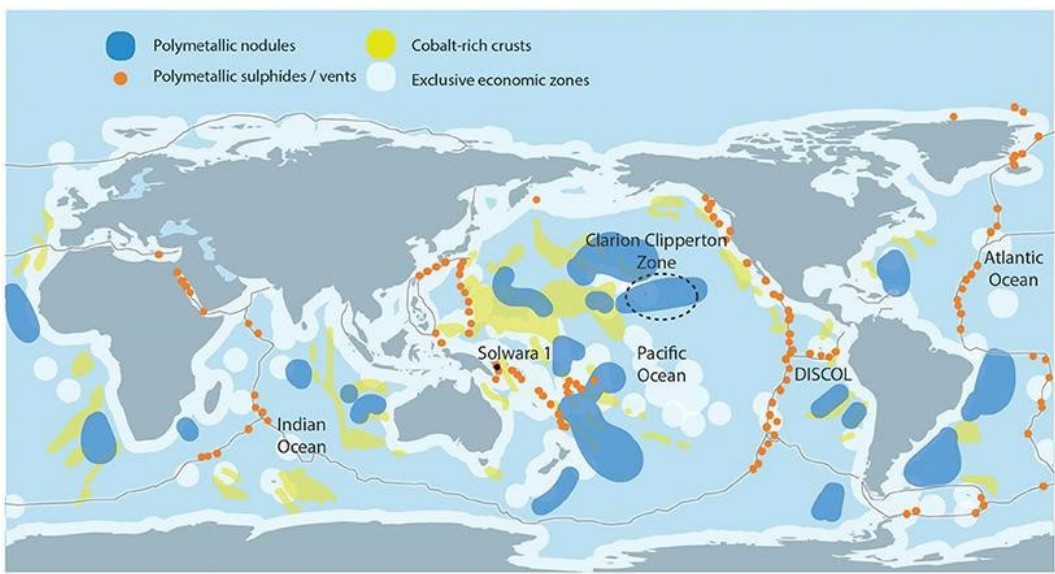

**Figure 2.** Deep-sea mining target minerals' distribution around the world [17].

### 2.2. Deep-Sea Mining Sustainability

A general definition of sustainability was given by United Nations [18] as '*meeting the needs of the present without compromising the ability of future generations to meet their own needs*'. The essence of sustainability research can be summarized as '*the integration of environmental health, social equity and economic vitality in order to create thriving, healthy, diverse and resilient communities for this generation and generations to come. The practice of sustainability recognizes how these issues are interconnected and requires a systems approach and an acknowledgement of complexity*' [19]. From the initial focus on environmental pollution to the current comprehensive interdisciplinary research, the concept of sustainable development is evolving over time. Ma [1] summarized a definition for deep-sea mining sustainability as: '*The sustainability applied in this thesis on DSM transport plans is a comprehensive concept connecting the 'environmental sustainability', 'economic sustainability', 'biological sustainability', 'energy use sustainability'. The sustainability research of DSM transport plan is to assess different DSM designs taking into consideration the technological, economic, environmental and social aspects simultaneously and find a compromise or an optimal balance among all influencing aspects.*'

The impacts on the deep-sea environment and ecology are the major concerns of scholars and environmentalists regarding the potential risks of deep-sea mining. Among them, biological resources, including animals, plants and microorganisms [20], have received great attention and research as they could be used as impact indicators for ecological damage [21]. Deep-sea mining biological impacts have various forms, for instance: (I) Suspended particles blocking the breathing system of seabed organisms. (II) Low-frequency noise and vibration hindering the communication of seabed organisms (such as hunting,

courtship). (III) High concentrations of toxic heavy metals affecting the growth, maturation cycle and spatial distribution of organisms. (IV) Redeposition of suspended particles and tailings have the potential to bury benthic flora and fauna, which can be fatal to these organisms. (V) The change in the total organic carbon content and seawater-dissolved oxygen may affect species' diversity, quantity and total biomass within the nearby water columns. (VI) Physical perturbation of seafloor habitats leads to the immediate death of static vegetation and animals with limited mobility. (VII) Light pollution from deep-sea mining operations can interfere with the role of weak light in deep-sea ecological communities. (VIII) Deep-sea mining activities may destroy undiscovered organisms and unknown substances, and these unknowns may determine human scientific and technological progress and the treatment of diseases in the future. The global geological hazards, climate change effects and ocean pollution caused by deep-sea mining activities are full of unknowns. In addition to the biological impacts, other environmental impacts consist of the physical destruction of benthic habitat, sediment plumes, geochemical impact, changes to total organic carbon content, oxygen penetration depth, pore water, and tailings, greenhouse gas emissions, toxic gas emissions, and noise–light pollution. All sub-environmental impacts are interconnected, which is reflected in the subsea biological impacts [5,6,15,22,23].

## 3. Life Cycle Assessment of a Deep-Sea Mining Project

As an upcoming emerging industry, deep-sea mining has great controversy in academia and in industry. Some environmentalists, in particular, regard deep-sea mining as a huge threat to deep-sea ecology and the global environment. On the other hand, as we all know, the huge ore reserves in the deep ocean can directly solve the problem of the world's ore resource crisis. Thus, the world should look at deep-sea mining from an objective, fair, and overall perspective [24–27]. Regarding deep-sea mining, the more urgent task is to scientifically analyze the environmental impact of its entire life cycle, the caused global and local problems, and also the research on environmental ecological restoration. The environmental impacts of deep-sea mining need to be compared not only with terrestrial mining, but also with certain marine natural disasters (such as the Tonga volcano eruption). In this way, people could obtain a more comprehensive and intuitive understanding of deep-sea mining, allowing us to truly understand the relatively green and environmentally friendly mining modes [28–31].

To obtain a relative objective comment on deep-sea mining, life cycle assessment could be utilized to analyze the environmental impact within the whole value chain from raw mineral ore mining, processing, transporting, transferring to alloys with different properties, manufacturing different kinds of products (e.g., wind turbine blades and electrical car batteries), and the final stage of material recycling and disposal [32,33]. The Metals Company compares terrestrial mining and seabed mining in terms of the perspective of mining life cycle assessment [34]. It compares four stages: the prospecting–exploration stage, the development stage, the mining and extraction stage, and the closure and reclamation stage (see Table 1). The comparison shows that the time required in the prospecting–exploration stage for the development of a deep-sea mining project is much shorter than for a terrestrial mining project, while the operating environment of deep-sea mining is more complex in the deep ocean, and the average project life is 20–30 years. As we all know, terrestrial mining itself is also a polluting industry, and is carried out around the world. The importance of the research is to prove either deep-sea mining or terrestrial mining as more sustainable for the environment.

**Table 1.** Mining life cycle cost comparison between deep-sea mining projects and terrestrial mining projects [34].

| Stage | Deep-Sea Mining | Terrestrial Mining |
| --- | --- | --- |
| **Prospecting and Exploration** | USD 20 million <br> <1~2 years <br> Non-invasive, simple; can do parallel with development | USD 1 million to USD 10 million dollars <br> 2~8 years <br> Locate economically viable ore deposits |

**Table 1.** *Cont.*

| Stage | Deep-Sea Mining | Terrestrial Mining |
|---|---|---|
| Development | USD 1 billion dollars to manufacture (capital cost)<br>4~6 years from discovery, including 3 years for environmental impact assessment and 2 years to construct and deploy | Up to billions of dollars (capital cost)<br>5~10 years<br>Plan and execute on building mines and supporting facilities |
| Mining and Extraction | <USD 1 billion per year (operation cost)<br>20~30 years<br>Ongoing collection operations at sea | Hundreds of millions to billions of dollars per year (operation cost)<br>5~50 years<br>Ongoing mining operation |
| Closure and Reclamation | Investigate ways of offset displacement of sea life and attachment surfaces | Restore the lands to the extent possible. Remove bridges, roads, cover ground and tailings ponds |

Notes: (1) The data of Table 1 is referenced from The Metals Company's white paper [34]. (2) The stages of prospecting and exploration, development, mining and extraction, closure and reclamation are referenced from Superfund Research Centre, The University of Arizona. (3) The statistical data for terrestrial mining are from the mining sites of U.S. Southwest areas. (4) The assessment data for deep-sea mining are from Deep Green Metals Inc.'s technical report summary prepared by AMC Consultants Pty Ltd., Melbourne, Australia, for the Northeast Pacific Ocean CCZ area's manganese nodule mines at the depth of 3800–4200 m.

Figure 3 describes the full life cycle assessment of a deep-sea mining project [33,35]. The procedure of life cycle assessment method application can be divided into four stages: goal and scope, data inventory, impact assessment, and interpretation–conclusion–recommendation (see Figure 4). In this paper, the life cycle starts from the raw material's exploitation, then proceeds to material processing, product manufacturing, distribution, and product use, and the last stage is product recycle–reuse–disposal. The raw material collection happening on the seabed is carried out by seafloor mining vehicles. The mineral processing consists of preliminary processing on the production support vessel and further processing on the land-based processing plants. Then, these seabed minerals are made into high carbon ferromanganese, disinfectant, electroplating materials, catalysts, and the other types of products. According to their different characteristics, these products can be used in electrical batteries, medical instruments, wind turbine blades, and the aerospace and defense industries, such as for tank casings, etc. As the age of the product usage increases, these products will eventually be recycled or disposed of [5,12,23,32–35]. The technologies involved in the life cycle consist of seabed mineral collecting, sediment cutting, tailing disposal, slurry processing, ore metallurgy, commodity manufacturing, shipment, retired product reuse, recycling, and degradation. The related economic costs, environmental pressures and subsequent reuse of recovered resources brought about by these technologies will also be the focus of our future quantitative analysis of deep-sea mining ore application. Analyzing the flow chart of the entire life cycle, it is obvious that deep-sea mining is a polluting industry when the raw material mining, material processing, and distribution stages are considered. However, at the same time, these rare metals can also be used in clean energy production processes, such as wind power, in electric vehicles, and in the solar industry. Therefore, whether a deep-sea mining project is environmental or not should be determined by the trade-off between the generated pollution and the environmental improvement made by the clean energy application [36–38].

Figure 5 describes the life cycle impact assessment criteria, including climate change, sediment plume, habitat physical destruction, toxic substances, benthic fauna and flora change, heavy metal concentration change, dissolved oxygen concentration change, sediment pore water change, noise and vibration influence, light pollution, tailing disposal, etc. These aspects could be used as environmental assessment indicators to present the pollution intensity of deep-sea mining.

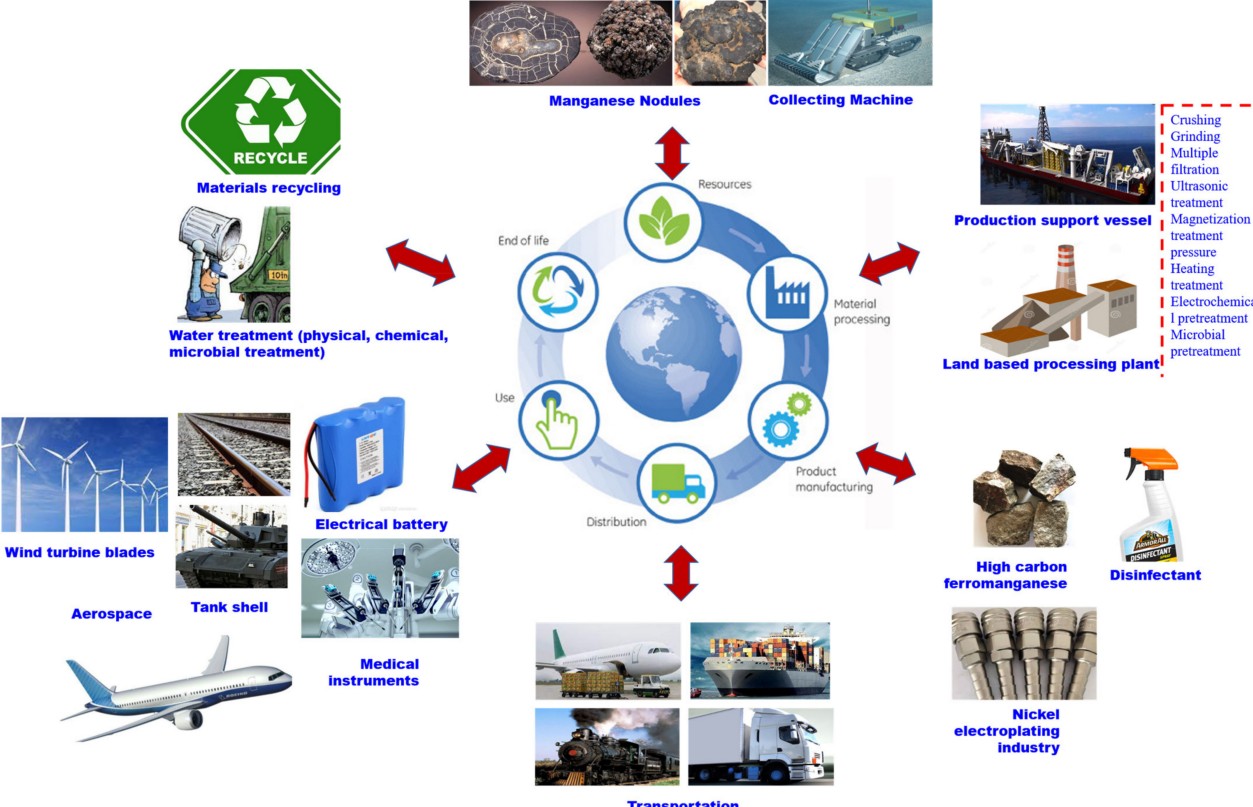

**Figure 3.** Full life cycle assessment of deep-sea mining project [33,35].

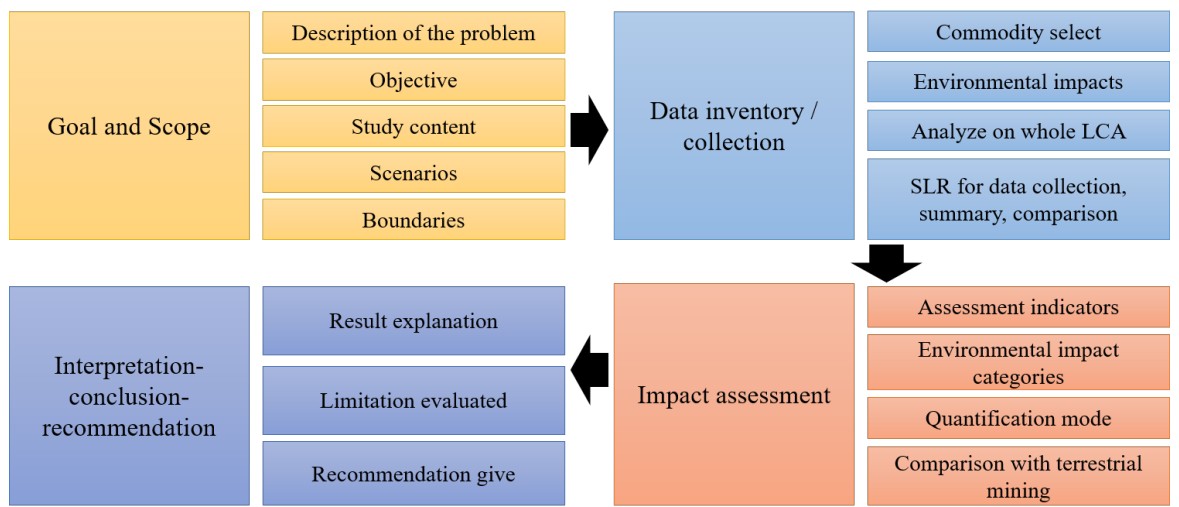

**Figure 4.** Life cycle assessment method application procedure [39].



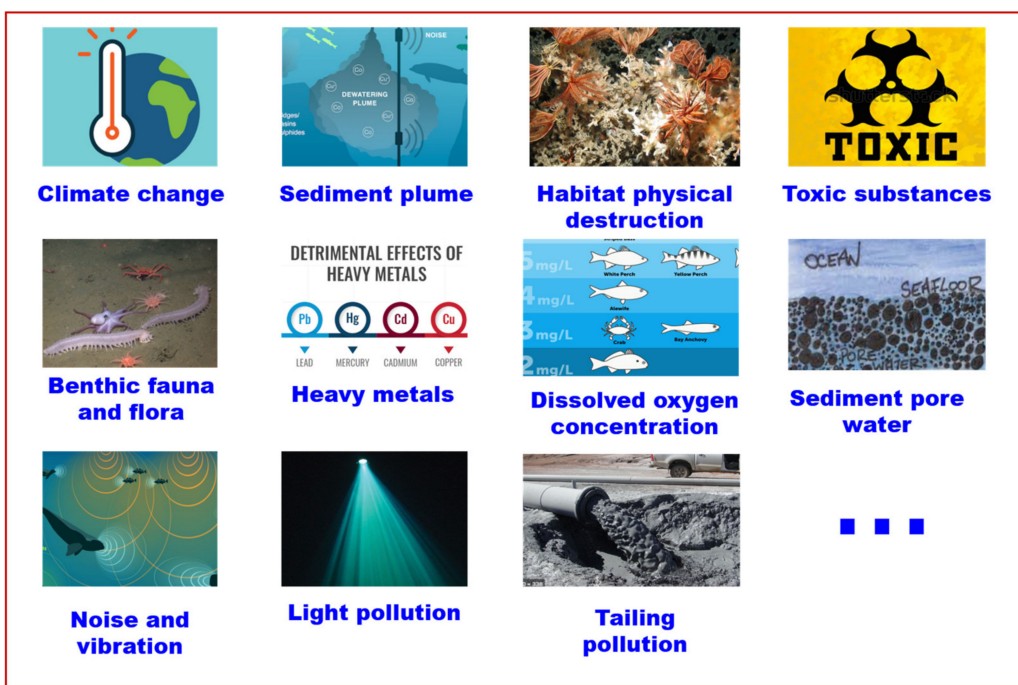

**Figure 5.** Life cycle impact assessment of deep-sea mining project [23,32].

## 4. Sustainability Development in Deep-Sea Mining

Sharma [40] claimed the sustainable development of deep-sea mining should consider economic, technical, technological, and environmental issues. Under several scenarios, Sharma quantitatively analyzed the seabed physical disturbance rate, overall mining efficiency, waste treatment, and impact of mining on the environment. Glover et al. [11] affirmed that deep-sea mining sustainability research prioritizes the emerging '*blue economy*' condition. Sustainability research should consider both the economic benefit and natural benefit to people. The research team also developed a taxon-focused method for deep ocean conservation that '*includes regulatory oversight to set targets for the delivery of taxonomic data*'. Santos et al. [41] coupled frontier technology development with a hazard assessment to address the major challenge of deep-sea mining's sustainable framework. This research intends to deepen humans' understanding of the deep sea through different kinds of new technologies, so as to promote the rational and green mining mode of seabed ore resources [41]. Kakee [42] and Childs [43] analyzed the urgency of environmental legislation combined with the sustainable deep-sea mining working mode. Vatalis et al. [29] qualitatively analyze how to obtain a sustainable deep-sea mining model from the perspective of the overall situation. The research emphasizes the need for deep-sea environmental impact assessments to consider the key factors of assessment procedure, laws and regulations, and interconnections with biodiversity. Carver et al. [44] emphasized the social, cultural and political dimensions of the development of deep-sea mining sustainability. Roche and Bice [9] also analyzed the social and community impact of deep-sea mining development.

The sustainability of deep-sea mining is not only restricted in the ecological and environmental field [9,11,29,40–44]. Current academic research on the sustainable development of deep-sea mining covers technological, economic, environmental, and social aspects (see Figure 6). Deep-sea mining impact will influence the atmosphere, land-based plant surroundings, the benthic seabed, and water columns; the specific environment indicators are shown in Figure 5 [5,23,32,45,46]. The economic aspect consists of the initial capital cost, operation and maintenance cost, and investment payback period [47–49]. The technological aspect is related to seabed mining vehicle technology, slurry lifting technology, production support vessel technology, and bulk carrier transport technology [50,51]. The social aspect

includes local community identity, employment, policies, and social infrastructure and services [8,9,44,52]. Economic, technological, environmental, and social impacts are all significant; it is meaningless to study one of them alone for the sustainable development of deep-sea mining. We should focus more energy and time on research into the quantitative relationship between these factors.

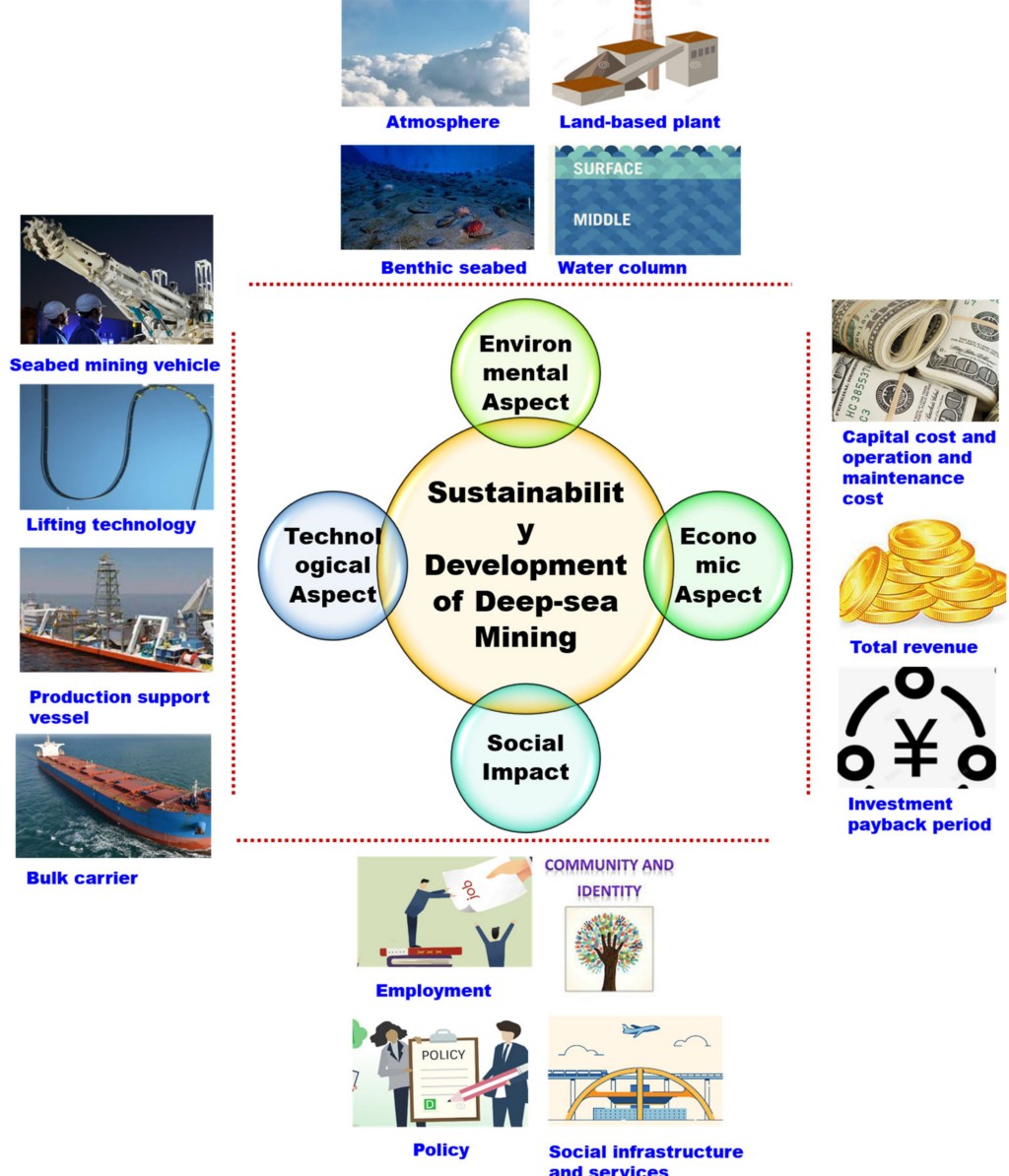

**Figure 6.** Sustainable development of deep-sea mining.

Table 2 lists the deep-sea mining sustainable development publications within the past ten years. We consulted the '*Web of Science*', '*Google Scholar*' and '*Scopus*' databases to review the literature related to deep-sea mining sustainability. The keywords that were used in our search were '*deep-sea mining*', '*sustainability development*', '*sustainability assessment*', and '*deep-sea resources exploitation*'. Based on the keyword searches, roughly 90 publications were identified, and among those, 30 publications were finally chosen (exclusion criteria included: publication was too old, not consistent with the research purpose, not a formal publication from a journal or conference, language not in English) to contribute to this article's discussion about useful research methods, definitions, frameworks, structures, and data.

**Table 2.** Summary of past 10 years' publications on sustainable development of deep-sea mining.

| Deep-Sea Mining Sustainable Development Components | Publications | Methods | Results, Comments and Suggestions |
|---|---|---|---|
| Social Ecological Technological Economic Governance | [8] | Mineral resources; Landscape method; Qualitative analysis. | Draw a map of stakeholder concerns for deep-sea mining in Australia; Dematerialization and recycling are underrated. |
| Social Environmental Economic | [9] | Comparison with terrestrial mining; Life cycle assessment; Qualitative analysis. | It may lead to employment competitions, economy and work practices; Increase the individual's awareness of human rights; Social–environmental impacts should be solved prioritized. |
| Social Technological Environmental Economic | [10] | Comparison with terrestrial mining activities; Qualitative analysis, | The EIA should cover the whole life cycle; Reinforce the mining procedure; Laws and regulations need to improved. |
| Economic Biological | [11] | Taxon-focused approach; Qualitative analysis. | Sustainable deep-sea mining development should not only rely on the modern ecosystem-based management approach. |
| Social Economic Environmental | [12] | Literature review; Qualitative analysis. | The application of circular economy for deep-sea mining exploitation would achieve many benefits for regulatory, technological and environmental improvement. |
| Social Economic Legal Governance | [28] | Qualitative analysis; Case analysis. | Deep-sea mining is an emerging activity; It lacks sufficient human, material resources; Monitoring system is necessary to ensure compliance; Research on the environmental impact is not sufficient. |
| Technological Economic Ecological | [53] | Expert-driven systematic conservation planning; Geospatial analysis; Expert opinion. | Establishment of marine protected areas; Biological and ecological act as the deep-sea mining impact indicators. |
| Social Environmental Engineering Management | [54] | Circular economy concept; Qualitative analysis. | Comparison of concept application of circular economy, environmental science, and sustainable development. |
| Social Economic Environmental | [55] | Deep-sea observing; Qualitative analysis. | Advocating the application of deep-sea observing method to obtain a sustainable deep ocean exploitation mode, |
| Technological Environmental Social | [56] | UN Sustainable development goals as the research direction; Qualitative analysis. | Research of challenges facing future mineral supply; Emphasis on the mineral recycling industry. |

**Table 2.** *Cont.*

| Deep-Sea Mining Sustainable Development Components | Publications | Methods | Results, Comments and Suggestions |
|---|---|---|---|
| Biological Economic International governance | [57] | Sustainability development of deep-sea fisheries; Comparison of fish data with economic drivers and governance contexts; Qualitative analysis. | Deep-sea commercial fishing has not been realized yet. |
| Social Governance Environmental | [58] | Research of dynamics of changes for mining sustainability development. | Progress for sustainability is being made; however, reform is still needed. |
| Social Governance | [59] | Description and assessment of key governance and institutional arrangements for social license to operation; Comprehensive literature review; Qualitative analysis. | Emphasis on the importance of social and local community in the mining activity; Social license for operation is just a start which needs further analysis. |
| Economic Biological Governance | [60] | Associated mitigation hierarchy method; Qualitative analysis. | Sustainability should not only consider the benefits of the current generation, but also future generations; The biodiversity loss due to deep-sea mining is poorly understood. |
| Environmental Legal Economic Societal | [61] | Comprehensive literature review; Qualitative analysis. | Comparison with terrestrial mining; Identify the current deep-sea mining sustainability research gaps; Highlighting the importance of interdisciplinary research. |
| Economic Environmental Technological | [62] | Literature review; Qualitative analysis; Comparison with terrestrial mining activities. | Discussion of the rare-earth element demand and renewed importance of deep-sea mining. |
| Social Economic Political Legal Environmental | [63] | Literature review; Qualitative analysis. | Environmental impact is researched at 'center stage' in deep-sea mining sustainability development. |
| Technological Societal Social–environmental | [64] | Literature review; Control, care, and conviviality application for sustainable development; Qualitative analysis. | 'Situated understandings of the interplay between control, care, and conviviality can help realize sustainability that does not reproduce the centralizing, control driven logic of Modern technocratic development' [64] |
| Environmental Technological Economic | [1,65,66] | Quantitative analysis; Advection–diffusion model; Deep-sea mining benefit calculation model, etc. | Deep-sea mining sustainable development should consider the technological, environmental and economic coupling relationship to obtain an objective assessment index. |
| Environmental Social Governance | [67] | Literature review; Spatial overlay approach; Qualitative analysis. | Compared to deep-sea mining technological research, that of environmental social and governance is less sufficient. |
| Economic Environmental Governance | [68] | Narrative literature review; Qualitative analysis. | Baseline data are lacking; Indicators in deep-sea mining sustainability are conflicting. |

**Table 2.** *Cont.*

| Deep-Sea Mining Sustainable Development Components | Publications | Methods | Results, Comments and Suggestions |
|---|---|---|---|
| Economic Political Governance Legal Ecological | [69] | Literature review; Qualitative analysis. | The sustainability indicators are various, and sometimes contradictory; A lot of still exist uncertainties. |
| Governance Legal Environmental Social | [70] | Qualitative comparison analysis. | Advocating collaboration of both international and national stakeholders; Advocating with regional and national academic institutions; Developing a long-term research program is necessary. |
| Social Environmental Technological Governance | [71] | Qualitative analysis. | 'There are significant, but not insurmountable, challenges to overcome before the deep-sea mining industry is recognized as economically viable or as a sustainable industry that can make a positive contribution to Pacific Island communities' [71] |
| Social Ecological | [72] | Qualitative analysis. | 'Assess whether the applicable legal frameworks at different levels attach sufficient importance to these traditional dimensions and to the human and societal aspects of seabed (mineral) resource management'; 'Identify best practices and formulate recommendations with regard to the current regulatory frameworks and seabed resource management approaches'. |

Note: Deep-Sea Mining Sustainability Development Components: the information in this column indicates which sustainability assessment parameters the paper addresses, such as environmental, ecological, technical, legal, and economic aspects.

Analyzing Table 2 shows that the application of sustainable development in other fields, such as terrestrial mining and oil and gas industries, has been well established [24,25,30]. However, the sustainable development of deep-sea mining still presents many uncertainties and research gaps [1,8–12,53–71].

- **Rare quantitative studies on the sustainable development of deep-sea mining**

Most of the theories and models applied to the sustainable development of deep-sea mining are extended from related industries [24,25,30,31]. This may also be because, officially, until now, deep-sea mining has not been industrialized. Moreover, sustainable development research is also a subject with a wide range of disciplines, involving technology, economy, the environment, and social ecology [40]. Many calculation and simulation results cannot be verified. Therefore, most of the research on the sustainable development of deep-sea mining is qualitative analysis.

- **Not sufficient research on environmental baseline data**

Environmental impact is a major component of the sustainable development of deep-sea mining. Environmental baseline data collection and monitoring determines the success of the entire project from the very beginning [46]. In the United Nations Convention of the Law of the Sea Article 145–196 and ISA deep-sea mining exploration code Part IV

Regulation 28, deep-ocean environmental baseline data measurement and monitoring is repeated more than once to emphasize their important research significance [73,74]. In 2021, the deep-sea mining research team from Linkoping University gave a narrative review on deep-sea mining activities. The research concluded that, currently, the deep ocean environmental baseline data are still lacking [68]. It is very difficult to complete a comprehensive and scientific environmental impact assessment report on deep-sea mining. It is precisely because of this that this document is highly valued by the International Seabed Authority and international environmental protection institutions (mostly NGOs) [23,32].

- **Missing relationship coupling research among these assessment indicators**

To obtain a comprehensive and scientific sustainability degree of deep-sea mining, the first priority is to find a series of representative assessment indicators. Some of these indicators are independent, and some of them have contradictory relationships [69]. For instance, sediment plume is one of the major environmental impact indicators which poses a great threat to seafloor life. The source of a sediment plume could be physical collection and cutting operation by a seafloor mining vehicle. It could also be submarine tailing disposal from the production support vessel. The leakage of the vertical lifting pipe system is another sediment plume source. Therefore, how to find a series of representative indicators while avoiding the repeated consideration of certain factors is a problem that needs to be solved urgently.

- **Deep-sea mining commercialization**

The locations of deep-sea mining can be divided into exclusive economic development zones and international seabed areas. The international sea area is not only managed by the International Seabed Authority, but is also subject to the constraints of international conventions such as the International Convention on the Law of the Sea. The technology of deep-sea mining is relatively mature, and now the biggest threshold restricting the industrialization of deep-sea mining is environmental pollution [75–77]. Many Pacific island countries do not have complete environmental and technical regulations to regulate deep-sea mining activities, and it is easy to issue deep-sea mining licenses in their own exclusive economic zones under the temptation of economic interests. This also brings trouble to neighboring countries, such as New Zealand and Australia. Because the environmental impact of deep-sea mining will never be limited only within the disturbed sea area, its direct environmental impact may be dozens of times the disturbed area of the seabed, and even cause potential global environmental pollution [78–82]. Therefore, the industrial exploitation of deep-sea mining is more likely to be carried out in the exclusive economic zone of less developed countries [83–85].

- **Cumulative environmental impacts**

Cumulative impact research is another way to obtain a comprehensive environmental impact assessment. This concept represents more than just the superimposed environmental pressures brought about by deep-sea mining activities over time and space. It also represents the environmental pressure after superimposed coupling between different sub-environmental impacts of deep-sea mining is considered. Clark et al. [46] summarizes three key elements when implementing a deep-sea mining cumulative impact assessment: *'1) Multiple sources of impact (either different types of mining operation, or different sectors such as fishing); 2) Additive or interactive processes (repetition leading to accumulation of impacts); 3) Different types of cumulative effects'*. A systematic literature review shows that, currently, there is no quantitative research to solve the deep-sea mining cumulative environmental impact problem. Smit and Spaling [22] categorized the research methodologies of cumulative environmental impact as analytical methods and planning methods (see Table 3). These methods could also be applied in deep-sea mining cumulative environmental impact analysis in the future.

**Table 3.** Cumulative environmental impact research method category [22].

| Analytical Methods | | Planning Methods | |
|---|---|---|---|
| **Category** | **Main Features** | **Category** | **Main Features** |
| Spatial analysis | Map spatial changes over time | Multi-criteria evaluation | Use of a priori criteria to evaluate alternatives |
| Network analysis | Identify the core structure and interactions of a system | Programming models | Optimize alternative objective functions subject to specified constraints |
| Biogeographic analysis | Analyze structure and function of landscape unit | Land suitability evaluation | Use ecological criteria to specify location and intensity of potential land uses |
| Interactive analysis | Sum additive and interactive effects, and identify higher-order effects | Process guidelines | Logic framework to conduct CEA |
| Ecological modelling | Model behavior of an environmental system or system components | — | — |
| Expert opinion | Problem solving using professional expertise | — | — |

- **Resource recycling**

  The total amount of resources on the earth is limited, but the consumption of human economic development increases year by year. Researchers have proposed to meet part of the demand for mineral resources through recycling [86–88]. One of the limitations of this technology is that the recycling rate of rare metal resources is very low. A lot of resources are disposed of as garbage due to backward technology [89], although this state has gradually changed, driven by the shortage of resources and the high price of minerals. Research scholars are paying attention to many rare metals, such as dysprosium and neodymium, to realize the objective of recycled metals to meet the demand of the future consumption [86]. Compared with the first-time exploitation of natural resources, the environmental impact of resource recycling is smaller and more controllable [90]. Therefore, the recycling of rare metals is considered to be one of the most feasible and environmentally friendly methods to solve the resource shortage crisis. Scholars estimate that in the next few decades, precious metal recycling is very likely to meet future demand [86–92].

- **Accepted environmental impact intensity analysis**

  Current academic circles have completed a lot of research in the field of the environmental impact of deep-sea mining, but there are also many small problems. One of the most obvious problems is that there are few studies quantifying the scope of acceptable environmental impacts of deep-sea mining. However, this research is the most pressing issue for all deep-sea mining stakeholders at present [1,4,21,46,84]. Currently, the International Seabed Authority only issues a series of seabed mineral exploration regulations and draft exploitation codes, which only list the definition, scope, stakeholders, and recommendations for environmental protection. If a deep-sea mining multinational company cannot provide a comprehensive and scientific quantitative analysis report on environmental impact according to the specific mining location and deep-sea mining technology, and quantitatively give the acceptable impact range of the deep-sea environment, it would be difficult for the International Seabed Authority to issue a license for the exploitation of seabed minerals within the international seabed area [68,93,94].

- **Environment recovery research**

  Thus far, deep-sea mining is still in the stage of academic research and resource exploration. Subsequent environmental restoration (after deep-sea mining operations) studies should also be properly arranged in advance [60,68,95–97]. Based on the systematic literature review analysis, all research in this field is qualitative and there is no experimental

research or quantitative analysis to solve this issue. With reference to the environmental pollution research of terrestrial mining and the environment restoration of deep-sea oil and gas exploitation, it is applicable to implement the artificial interventions such as physical precipitation and flocculation methods, electrochemical precipitation methods and microbial accelerated decomposition methods to accelerate the recovery process of the deep-sea environment [60,68,95–97].

## 5. Conclusions

Although the study of deep-sea mining has a long history, there are still many restrictions and limitations on its industrial exploitation due to unknown environmental baseline and monitoring data, ecological–social impact, and the uncertainty of environmental threats posed by the proposed technologies. The life cycle assessment approach is discussed here to analyze deep-sea mining sustainability in comparison with terrestrial mining activities. In future research, the author will apply the whole life cycle assessment method to quantify and compare the environmental pollution caused by deep-sea mining and terrestrial mining to evaluate which mining mode is more environmentally friendly. The main purpose of this article is to discuss the research status of the sustainable development of deep-sea mining and summarize the existing research gaps, including the lack of environmental baseline data, environmental data detection systems and equipment, quantitative research on cumulative environmental impact, and lack of analysis and research on the acceptable degree of environmental pollution. The significance of this paper is to clarify the research status and research gaps of deep-sea mining in the field of sustainable development research. It is also hoped that, through this article, the academic community can adjust their direction of research appropriately and promote the industrialization of deep-sea mining in an efficient manner.

**Author Contributions:** W.M. wrote the draft; Y.S., K.Z., X.L. and Y.D. reviewed the article. All authors have read and agreed to the published version of the manuscript.

**Funding:** This work was supported by Hainan University High-Level Talents Research Start-Up Funding (to W.M.) [grant number KYQD(ZR)-22060], the International Association of Maritime Universities [research project number 20220304], and the National Fund Committee Key Project [research project number 52231012].

**Institutional Review Board Statement:** Not applicable.

**Informed Consent Statement:** Not applicable.

**Conflicts of Interest:** The authors declare no conflict of interest.

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
