# Peer review of "Status of Sustainability Development of Deep-Sea Mining Activities"

_jmse, doi:10.3390/jmse10101508_

Round 1

Reviewer 1 Report (Previous Reviewer 1)

It seems that a more complete version of this article has been prepared and can be considered for publication in this journal. 

Author Response

Dear Reviewer 1:

Thank very much for your comments. Please have a check of my responses and revision as follows.

Comment: It seems that a more complete version of this article has been prepared and can be considered for publication in this journal. 

Response: Thanks for your valuable and professional comments improving this paper. 

Kind regards,

Wenbin Ma

Reviewer 2 Report (Previous Reviewer 3)

Please see reviewer's report.

Author Response

Dear Reviewer 2,

Thanks for your valuable and professional comments improving this paper. 

Kind regards,

Authors

Reviewer 3 Report (Previous Reviewer 4)

The paper is acceptable for publication.

Author Response

Dear Reviewer 3:

Thank very much for your comments. Please have a check of my responses and revision as follows.

Comment: The paper is acceptable for publication.

Response: Thanks for your valuable and professional comments improving this paper. 

Kind regards,

Wenbin Ma

Round 2

Reviewer 2 Report (Previous Reviewer 3)

Please see reviewer's report.

Author Response

Dear Reviewer,

Thanks for your comments improving this paper.

      Comment 1: The reference citation numbers in square brackets look like superscript. Please check the instructions.

      Response: Thanks for your comments. The reference citation is revised accordingly in the manuscript.

Kind regards,

Authors

This manuscript is a resubmission of an earlier submission. The following is a list of the peer review reports and author responses from that submission.

Round 1

Reviewer 1 Report

Economic opportunities for sea mining have created a lot of incentives for research and development in this area. However, there is still a long way to go to achieve sustainable development regarding mineral activities in the depths of the seas. The draft is also an overview of this. However, the main problems and questions for it must be solved in this way.

Some Figures, such as Figure 6, do not have references. Referenced figures must have written permission from the publisher.

Written scientific English language is needed to be overall corrected.

Example: Roche and Bice (2013) also analyzed the anticipated social and community impact on deep-sea mining development.

Some titles are too long:

Example:

Parts of indicators are not independent, even contradictory. The couple research all these indicators together and get a comprehensive assessment index is missing.

Some titles are not suitable.

Example

Environmental baseline data is lacking

A series of deep-sea Mining problems have been generalized without a practical method or policy for future development, such as a quantitative measure of environmental impact with appropriate references. 

Author Response

Dear Reviewer 1:

Thank very much for your comments. Please have a check of my responses and revision as follows.

Comment 1: Economic opportunities for sea mining have created a lot of incentives for research and development in this area. However, there is still a long way to go to achieve sustainable development regarding mineral activities in the depths of the seas. The draft is also an overview of this. However, the main problems and questions for it must be solved in this way. 

Response: Thanks for your comments. Firstly, I am totally agreed with you that the economic profit is the major motivation for the R&D of deep-sea mining. It also includes the reasons for the shortage of terrestrial mineral resources and the apparent decline in the quality of terrestrial ore. In addition, many countries (such as China) have a large demand for ores, and the high cost of importing ores has also increased their interest in deep-sea mining. It is as shown in Chapter 2 Lines 78-82. Secondly, I am also totally agreed with you that the sustainability development of deep-sea mining still has a long way to go. Many research institutions hold different views on deep sea mining, which also determines their research direction. For example, most of the research on deep sea mining by China Ocean Association and China Minmetals Corporation is the technology required for industrialized mining, such as lift pump system, vertical pipeline design and manufacture, seabed mining vehicle and mining vessel design, ignoring the protection and restoration of the deep-sea environment. Through this article, I also hope that the research on deep sea mining will be as comprehensive and systematic as possible, and to evaluate the feasibility and sustainability of seabed mining from a full life cycle perspective. We also recommend the research of deep-sea mining sustainability should not only limited in the aspect of environmental impact, which should be considered integrating technological, economic, environmental and social aspects simultaneously and find a compromise balance or an optimal balance between all influencing aspects. It is shown in Section 2.2 Lines 100-105.   

Comment 2: Some Figures, such as Figure 6, do not have references. Referenced figures must have written permission from the publisher.

Response: Thanks for your comments. Fig. 6 is an original image to explain the key points in the sustainability development of deep-sea mining.

Comment 3: Written scientific English language is needed to be overall corrected.

Response: Thanks for your comments. English language in this paper is corrected. Several correction examples are shown as follows (For more modification details, please see the original text):

On Page 1 Line 1-2, the title is changed as ‘Status of Sustainability Development of Deep-sea Mining Activities’.

On Page 1 Lines 14-15, the sentence is revised as ‘In the end, this paper summarizes the research gaps that exist in the sustainable development of deep-sea mining, including…’.

On Page 1 Lines 30-31, the sentence is revised as ‘The environmental pollution and sustainable development issues of deep-sea mining have become one of the biggest thresholds for its way forward’.

On Page 1 Lines 32-33, the sentence is revised as ‘Fig. 1 describes a schematic diagram of one typical deep-sea mining project’.

On Page 3 Lines 83-84, the sentence is revised as ‘Since the concept of deep-sea mining was first proposed, more than half a century later, it is still in the stage of exploration and experimental research’.

On Page 3 Lines 112-114, the following sentence is deleted ‘which would analyze its environmental and social impacts, economic profitability, production rate, working efficiency and social impacts’.

On Page 4 Lines 116-117, the sentence ‘find a compromise balance or an optimal balance between all influencing aspects’ is replaced by ‘find a compromise or an optimal balance among all influencing aspects’.

On Page 4 Line 136, ‘And’ is deleted.

On Page 5 Lines 158-161, the sentence is revised as ‘Regarding deep sea mining, the more urgent task is to scientifically analyze the environmental impact of its entire life cycle, the caused direct and indirect global and local problems, and also the research on environmental ecological restoration’.

On Page 5 Line 177, the phrase ‘both of them’ is added.

On Page 5 Line 179, the word ‘less’ is replaced by ‘shorter’.

On Page 6 Line 193, the punctuation is revised.

On Page 8 Line 225, the word ‘And’ is deleted.

On Page 9 Line 245, the word ‘topic’ is deleted.

On Page 14 Line 294, ‘land mining’ is replaced by ‘terrestrial mining’.

In Chapter 4 Lines 221-223, ‘Roche and Bice (2013) also analyzed the anticipated social and community impact on deep-sea mining development’ is revised as ‘Roche and Bice (2013) also analyzed the social and community assessment elements impact on deep-sea mining development’.

Comment 4: Some titles are too long: Example: Parts of indicators are not independent, even contradictory. The couple research all these indicators together and get a comprehensive assessment index is missing. And some titles are not suitable. Example: Environmental baseline data is lacking.

Response: Thanks for your comments. In Chapter 4 Lines 293-296, ‘Parts of indicators are not independent, even contradictory. The couple research all these indicators together and get a comprehensive assessment index is missing’ is changed as ‘Missing of relationship coupling research among these assessment indicators’. In Chapter 4 Line 279, ‘Environmental baseline data is lacking’ is changed as ‘Not sufficient research on environmental baseline data’.

Comment 5: A series of deep-sea Mining problems have been generalized without a practical method or policy for future development, such as a quantitative measure of environmental impact with appropriate references. 

Response: Thanks for your comments. As we all agreed, deep-sea mining R&D still has a long way to go to its industrialization. I agree with Reviewer 1 that if we want to carry out specific sustainable development research on deep-sea mining, it is necessary to use quantitative analysis, mathematic methods, concrete seabed mining policy, etc. Different research directions will analyze the advantages and disadvantages of deep-sea mining activities from different perspectives. For instance, authors tried to analyze the deep-sea mining environmental impact – sediment plume transport using advection-diffusion model, see the following reference. Ma, W., Schott, D. and van Rhee, C., 2019. Numerical calculations of environmental impacts for deep sea mining activities. Science of The Total Environment, 652, pp.996-1012. In the above publication, quantitative analysis is carried out. In this reviewed article, we are from a global perspective to analyze the sustainable development of deep-sea mining ore application process. In the next step, we plan to use the quantitative analysis to achieve the evaluation of sustainable development of deep-sea mining. The mathematical methods involved will include life cycle assessment method (cradle-to-cradle mode), the technical, environmental and economical modelling assemblages. This article analyzes the elements involved and the existing research loopholes in the research topic of sustainable development of deep-sea mining from the perspective of qualitative analysis. It can serve as a guiding document for our further in-depth quantitative research. 

Finally, thank you again for your valuable comments.

Kind regards,

Wenbin Ma

Reviewer 2 Report

The authors’ efforts introduced in this article are wonderful challenges. However, no actual venture has started deep-sea mining is the headache point. Because the one might start a mining venture from 2020 was for polymetallic sulfides, no realistic information is available for polymetallic nodules. Especially there are little information about the processing plant for polymetallic nodules. The four-metal recovery in a plant is quite different with conventional processing plants. The CAPEX and OPEX of processing plant may be a half of total but difficult to estimate. In addition, comparison of sustainability with land mining must cover two or three mining operations on land.

The revision points are marked in yellow in the attached PDF.

1.    Recommend to add “Status of” at the top of paper title.

2.    There are three targets for deep-sea mining, polymetallic nodules, polymetallic sulfides, cobalt-rich crusts as shown in Fig. 2. The target of this study is polymetallic nodules, because most of the references are related with polymetallic nodules. In case of polymetallic sulfides, because the same land-based processing plant will be used, the cost and environmental conditions are different with polymetallic nodules. The word usage of “deep-sea mining” in this study covers only polymetallic nodules or it covers polymetallic sulfides either should be clearly defined.

3.    The sentence in Lines 28-30 on P. 1 is not acceptable explanation. Without no reference, “the concept of deep-sea mining was proposed in 1976,” is mentioned.  Then “its commercialization has still not been realized (Mero, 1970)” is mentioned.  The time order is reverse and the authors don’t aware this strange situation at all. Mero mentioned a possibility of ferromanganese nodules as future metal supply source. The sentence structure must be started from Mero’s viewpoint. In addition, in 1976 already a few consortia finished their component experiments and they completed their pilot mining tests in 1978 and 1979. The concept proposal was about 10 years earlier. The following one might be an example.

Flipse, J.E. An engineering approach to ocean mining. Offshore Technology Conference, Houston, TX, USA, 18–21 May 1969.

4.    Which is an important reference, Mero (1970) or Mero (1965)? Usually Mero (1965) is selected, because it is the first one and earlier.

Mero, J.L. The Mineral Resources of the Sea; Oceanography Series, 1; Elsevier: Amsterdam, The Netherlands, 1965; p. 312.

5.    Change from Teresa (2022) to Kennedy (2022). Teresa is first name.

6.    The sentence in Lines 191-193 on P. 6 doesn’t describe an important point. Land mining itself is also a polluting industry. Therefore, comparison of polluting with land mining must be included. Of course, comparison of benefit with land mining depending on the metal contents must be considered. The most important point must be clarified is which is more harmful to environment the land mining or the deep-sea mining.

7.    Add the journal name and pages of Sharma (2011).

8.    Delete university names in lines 222 and 223 on P. 7.

9.    The description on lines 332-334 must be revised. This is an opinion by the authors and not popular in scientific, engineering, and industrial people.

10. Clarification of the word “cumulative environmental impacts” is necessary. Except deep-sea mining other impacts such as fishing, acoustic survey equipment, and vessel cruising already exist. First of all, the clarification of the cumulative impacts at current situation is necessary, Then, addition of multi-operations of deep-sea mining should be examined. The reference looks very old one.

11.  Revision is necessary on lines 378 and 379. ISA is an organization to manage the deep-sea mining in international area. The regulations for Environmental Assessment by ISA list many recommendations and few requirements. The ones foe Exploitation proposed by ISA for discussion do not include many strong requirements. 

Author Response

Dear Reviewer 2:

Thank very much for your comments. Please have a check of my responses and revision as follows.

Comment: The authors’ efforts introduced in this article are wonderful challenges. However, no actual venture has started deep-sea mining is the headache point. Because the one might start a mining venture from 2020 was for polymetallic sulfides, no realistic information is available for polymetallic nodules. Especially there are little information about the processing plant for polymetallic nodules. The four-metal recovery in a plant is quite different with conventional processing plants. The CAPEX and OPEX of processing plant may be a half of total but difficult to estimate. In addition, comparison of sustainability with land mining must cover two or three mining operations on land.

Response: Thanks for your comments. I am totally agreed with your opinions. Although the concept of deep-sea mining has been proposed for more than half a century, research institutions, companies and government departments around the world have not stopped its research and development, there are still many research gaps. Officially for this reason, the International Seabed Authority has not issued a mining certificate for the time being. In terms of mining difficulty, seabed manganese nodules are relatively small. The research on its environmental and biological effects is also not sufficient. Many environmental groups and NGOs really worry about it. It is precisely because the industrialized mining of deep-sea mining has not yet begun, and its downstream, including ore smelting and processing plants, are not fully equipped. Regarding the economic profitability of deep-sea mining, parts of researchers in the academic community have made some estimates (CAPEX and OPEX), of course, under a series of assumptions. Most of these studies show the enormous economic potential of deep-sea mining. Precisely how to find the optimal balance in environmental, economic, technological and social perspectives has not been resolved. In the author's eyes, research on sustainable development of deep-sea mining can be divided into two comparisons, that is, horizontal comparison of land mining, and vertical comparison of the difference in environmental pressures with or without waste recycling and retired products reuse.

The revision points are marked in yellow in the attached PDF.

Comment 1:  Recommend to add “Status of” at the top of paper title.

Response: Thanks for your comments. The manuscript title is changed as ‘Status of Sustainability Development Analysis of Deep-sea Mining Activities’.

Comment 2: There are three targets for deep-sea mining, polymetallic nodules, polymetallic sulfides, cobalt-rich crusts as shown in Fig. 2. The target of this study is polymetallic nodules, because most of the references are related with polymetallic nodules. In case of polymetallic sulfides, because the same land-based processing plant will be used, the cost and environmental conditions are different with polymetallic nodules. The word usage of “deep-sea mining” in this study covers only polymetallic nodules or it covers polymetallic sulfides either should be clearly defined.

Response: Thanks for your comments. This article majorly focuses on the polymetallic nodule mining activities. So, in Chapter 1 Lines 62-64, the following sentence is revised as ‘The objective of this paper is to analyze the issues existing in deep-sea mining (mainly focusing on polymetallic nodule mining activities) sustainability development’.   

Comment 3: The sentence in Lines 28-30 on P. 1 is not acceptable explanation. Without no reference, “the concept of deep-sea mining was proposed in 1976,” is mentioned.  Then “its commercialization has still not been realized (Mero, 1970)” is mentioned.  The time order is reverse and the authors don’t aware this strange situation at all. Mero mentioned a possibility of ferromanganese nodules as future metal supply source. The sentence structure must be started from Mero’s viewpoint. In addition, in 1976 already a few consortia finished their component experiments and they completed their pilot mining tests in 1978 and 1979. The concept proposal was about 10 years earlier. The following one might be an example. Flipse, J.E. An engineering approach to ocean mining. Offshore Technology Conference, Houston, TX, USA, 18–21 May 1969.

Response: Thanks for your comments. The timeline is clarified as follows. In Chapter 1 Page 1 Lines 28-30, the following sentence ‘Since the concept of deep-sea mining was proposed in 1976, its commercialization has still not been realized (Mero, 1970)’ is revised as ‘Since the concept of deep-sea mining was proposed in 1960s, its commercialization has still not been realized (Flipse, 1969; Mero, 1965)’.      

Comment 4: Which is an important reference, Mero (1970) or Mero (1965)? Usually Mero (1965) is selected, because it is the first one and earlier. Mero, J.L. The Mineral Resources of the Sea; Oceanography Series, 1; Elsevier: Amsterdam, The Netherlands, 1965; p. 312.

Response: Thanks for your comments. The reference should be Mero (1965), as you mentioned. So, in the whole paper, Mero (1970) is replaced by Mero (1965).   

Comment 5: Change from Teresa (2022) to Kennedy (2022). Teresa is first name.

Response: Thanks for your comments. Through the whole paper, Teresa (2022) is replaces by Kennedy (2022).   

Comment 6: The sentence in Lines 191-193 on P. 6 doesn’t describe an important point. Land mining itself is also a polluting industry. Therefore, comparison of polluting with land mining must be included. Of course, comparison of benefit with land mining depending on the metal contents must be considered. The most important point must be clarified is which is more harmful to environment the land mining or the deep-sea mining.

Response: Thanks for your comments. You gave me a very valuable suggestion. Therefore, in Chapter 3 Page 5 Lines 164-167, the following sentence is added to explain the comparison meaning between deep-sea mining and terrestrial mining: ‘As we all known, terrestrial mining itself is also a polluting industry, which is carried out around the world. The importance of the research is to prove either deep-sea mining or terrestrial mining is more harmful to the environment’.

Comment 7: Add the journal name and pages of Sharma (2011).

Response: Thanks for your comments. The missing information is added as follows: ‘Sharma, R., 2011. Deep-sea mining: Economic, technical, technological, and environmental considerations for sustainable development. Marine Technology Society Journal, 45: 28-41’.

Comment 8: Delete university names in lines 222 and 223 on P. 7.

Response: Thanks for your comments. The university names are deleted accordingly.

Comment 9: The description on lines 332-334 must be revised. This is an opinion by the authors and not popular in scientific, engineering, and industrial people.

Response: Thanks for your comment. For certain environmental impacts, both academia and industry have the same perception, which means that we all recognize these types of environmental impacts. It includes the sediment plume, seabed habitat disturbance, ocean species impact, and so on. A part of researchers also worried out the potential global environmental pollution. For instance, Hunter (2018) stated that ‘recent scientific research, however, has revealed that the deep seabed, and hydrothermal vents in particular, make potentially critical contributions to both biodiversity and global climate regulation’. From my point of view, there are some differences in the research points and research directions of academia, industry, and environmentalists on deep sea mining. In the article, the author only uses 'maybe' to illustrate the possible global environmental impacts of deep-sea mining.

Comment 10. Clarification of the word “cumulative environmental impacts” is necessary. Except deep-sea mining other impacts such as fishing, acoustic survey equipment, and vessel cruising already exist. First of all, the clarification of the cumulative impacts at current situation is necessary, Then, addition of multi-operations of deep-sea mining should be examined. The reference looks very old one.

Response: Thanks for your comment. Firstly, on Page 14 Lines 330-333, the cumulative environmental impact of deep-sea mining is clarified as follows: ‘This concept represents more than just the superimposed environmental pressures brought about by deep-sea mining activities over time and space. At the same time, it also represents the environmental pressure after superimposed coupling between different sub-environmental impacts of deep-sea mining’. Based on the literature review, there is not sufficient research to address this issue. There are many reasons for this phenomenon, one of which is the difficulty of reaching the deep-sea environment to conduct relevant experiments and collect the corresponding environmental baseline data. The second is that the environmental impact of deep-sea mining has a large time span. It is even more difficult to simulate the technical disturbance of deep-sea mining in the deep-sea environment in the early stage, and to continuously observe, collect and evaluate data in the next few years or even decades.

Comment 11. Revision is necessary on lines 378 and 379. ISA is an organization to manage the deep-sea mining in international area. The regulations for Environmental Assessment by ISA list many recommendations and few requirements. The ones foe Exploitation proposed by ISA for discussion do not include many strong requirements. 

Response: Thanks for your comments. I am totally agreed with your opinion. International seabed authority is responsible for the deep-sea mining activities within the international seabed area. Until now, this organization only issued the exploration regulation and draft exploitation codes. As you mentioned, these standards and codes does not include many strong requirements. With the deep-sea mining coming closer, it is much urgent for ISA to issue a regulation with quantitative standard, which would be used as a guideline (or limitation) to control the future seabed mining activities within a sustainable mode. For example, in land mining, the minimum concentration of mining ore, the discharge standard of waste water, and the disposal standard of waste are clearly stipulated. On Page 15 Lines 368-371, the following sentences are added: ‘Currently, International Seabed Authority only issued a series of seabed mineral exploration regulation and draft exploitation codes, which only list the definition, scope, stakeholders, recommendations of environmental protection’.

Finally, thank you again for your valuable comments.

Kind regards,

Wenbin Ma

Reviewer 3 Report

This paper is not acceptable because it contains no valuable results obtained from life cycle assessment for the environmental impacts of deep-sea mining although the authors state “this paper also attempts to use the full life cycle assessment method to analyze the environmental impact of the entire process of deep-sea mining ore application” (L12~14).

Author Response

Dear Reviewer 3:

Thank very much for your comments. Please have a check of my responses and revision as follows.

Comment: This paper is not acceptable because it contains no valuable results obtained from life cycle assessment for the environmental impacts of deep-sea mining although the authors state “this paper also attempts to use the full life cycle assessment method to analyze the environmental impact of the entire process of deep-sea mining ore application” (L12~14).

Response: Thanks for your comments. The objective of this paper is to analyze the research status of deep-sea mining (mainly focusing on polymetallic nodule mining activities) sustainability development. It attempts to use the full life cycle assessment method to analyze the environmental impact of the entire process of deep-sea mining ore application and summarizes the existing research gaps existing in this field. On Page 6 Lines 197-203, the ‘Analysing Fig. 3, it is obvious that deep-sea mining is a polluting industry if we only consider its raw material mining, material processing and distribution stages. While at the same time, these rare metal ores can in turn be used in clean energy production processes such as wind power, electric vehicles and the solar industry. Therefore, whether deep-sea mining project is environmental or not should be determined by the trade-off between the generated pollution and the environment improved by the clean energy application’. Life cycle assessment method application is the major content of section 3. In our opinion, the most valuable part of this article is the identification of loopholes in the sustainable development of deep-sea mining. These research gaps can serve as our research directions. Additionally, adopting the suggestions and comments from the other 6 reviewers, I am sure the quality of this article has been improved.

Besides the above comments, the English language is revised through the whole paper. Several correction examples are shown as follows (For more modification details, please see the original text):   

On Page 1 Line 1-2, the title is changed as ‘Status of Sustainability Development of Deep-sea Mining Activities’.

On Page 1 Lines 14-15, the sentence is revised as ‘In the end, this paper summarizes the research gaps that exist in the sustainable development of deep-sea mining, including…’.

On Page 1 Lines 30-31, the sentence is revised as ‘The environmental pollution and sustainable development issues of deep-sea mining have become one of the biggest thresholds for its way forward’.

On Page 1 Lines 32-33, the sentence is revised as ‘Fig. 1 describes a schematic diagram of one typical deep-sea mining project’.

On Page 3 Lines 83-84, the sentence is revised as ‘Since the concept of deep-sea mining was first proposed, more than half a century later, it is still in the stage of exploration and experimental research’.

On Page 3 Lines 112-114, the following sentence is deleted ‘which would analyze its environmental and social impacts, economic profitability, production rate, working efficiency and social impacts’.

On Page 4 Lines 116-117, the sentence ‘find a compromise balance or an optimal balance between all influencing aspects’ is replaced by ‘find a compromise or an optimal balance among all influencing aspects’.

On Page 4 Line 136, ‘And’ is deleted.

On Page 5 Lines 158-161, the sentence is revised as ‘Regarding deep sea mining, the more urgent task is to scientifically analyze the environmental impact of its entire life cycle, the caused direct and indirect global and local problems, and also the research on environmental ecological restoration’.

On Page 5 Line 177, the phrase ‘both of them’ is added.

On Page 5 Line 179, the word ‘less’ is replaced by ‘shorter’.

On Page 6 Line 193, the punctuation is revised.

On Page 8 Line 225, the word ‘And’ is deleted.

On Page 9 Line 245, the word ‘topic’ is deleted.

On Page 14 Line 294, ‘land mining’ is replaced by ‘terrestrial mining’.

In Chapter 4 Lines 221-223, ‘Roche and Bice (2013) also analyzed the anticipated social and community impact on deep-sea mining development’ is revised as ‘Roche and Bice (2013) also analyzed the social and community assessment elements impact on deep-sea mining development’.

Finally, thank you again for your valuable comments.

Kind regards,

Wenbin Ma

Reviewer 4 Report

The authors present a literature review of work done related to the sustainability of deep sea mining.  At the outset, it is important to note that I am a scientist, but not an expert in this field.  I am therefore am reading this article from the perspective of an interested observer who happens to have a technical background.

The topic of the work is very relevant and the authors appear to have done a thorough search of the literature (though again, I am not qualified to assess the completeness of the review).  I do have a couple of observations/suggestions prior to publication.

First, the authors should go through the article and revise statements that could be construed as politically motivated.  For example, the first two sentences of §3 are somewhat awkward.  It appears that the authors may simply be conveying a thematic message put forward in the articles they are citing.  But the way it is worded, it could also be interpreted as a value judgement on social and political perceptions of deep sea mining.  It is recommended that the authors avoid such linguistic complexities to ensure that the review is as objective as possible.

Second, it is not clear that Table 2 has any particular value.  This is, of course, an on-line journal.  So page length isn't really an issue.  But there doesn't appear to be any value added that isn't already in the text.

Author Response

Dear Reviewer 4:

Thank very much for your comments. Please have a check of my responses and revision as follows.

Comment 1: The authors present a literature review of work done related to the sustainability of deep-sea mining.  At the outset, it is important to note that I am a scientist, but not an expert in this field.  I am therefore am reading this article from the perspective of an interested observer who happens to have a technical background.

Response: Thanks for your comments. It is a great pleasure to communicate with you about the sustainable development of deep-sea mining. As an upcoming new industry, deep sea mining has great controversy in academia and industry. A part of environmentalists, in particular, regard deep-sea mining as a huge threat to deep-sea ecology and the global environment. But just like offshore oil and gas, land mining, and pelagic fishing, we cannot stop these activities entirely because of environmental pollution. Especially now that the contradiction between the rapid development of the world economy and the shortage of ore resources is further intensified. The more importance thing is to utilize the scientific method to approve these activities could be implemented within a sustainable degree. The objective of this paper is to analyze the research status of deep-sea mining (mainly focusing on polymetallic nodule mining activities) sustainability development. It attempts to use the full life cycle assessment method to analyze the environmental impact of the entire process of deep-sea mining ore application and summarizes the existing research gaps existing in this field.    

Comment 2: The topic of the work is very relevant and the authors appear to have done a thorough search of the literature (though again, I am not qualified to assess the completeness of the review).  I do have a couple of observations/suggestions prior to publication. First, the authors should go through the article and revise statements that could be construed as politically motivated. For example, the first two sentences of §3 are somewhat awkward. It appears that the authors may simply be conveying a thematic message put forward in the articles they are citing.  But the way it is worded, it could also be interpreted as a value judgement on social and political perceptions of deep-sea mining.  It is recommended that the authors avoid such linguistic complexities to ensure that the review is as objective as possible.

Response: Thanks for your comments. I am totally agreed with your opinion. So, on Page 4-5 Lines 150-156, the sentences ‘The world should look at deep-sea mining from an objective, fair and overall perspective. No one would deny the environmental hazards of deep-sea mining. However, just like the industries of marine oil extraction, marine gas extraction, trawling, hydroelectric power, nuclear power generation, etc., the world cannot decide to ban them because of their environmental damage’ is replaced by ‘As an upcoming emerging industry, deep-sea mining has great controversy in academia and industry. A part of environmentalists, in particular, regard deep-sea mining as a huge threat to deep-sea ecology and global environment. At the same time, as we all known, the huge ore reserves in the deep sea can directly solve the problem of the world's ore resource crisis. So, the world should look at deep-sea mining from an objective, fair and overall perspective’.

Comment 3: It is not clear that Table 2 has any particular value.  This is, of course, an on-line journal.  So page length isn't really an issue.  But there doesn't appear to be any value added that isn't already in the text.

Response: Thanks for your comments. I totally understand what you mean. Actually, before the manuscript submission, we had thought that maybe it is better to put table 2 into the supplement. Table 2 gives a summary of past 10 years publications on deep-sea mining sustainability development. In this table, it consists of the information of the deep-sea mining sustainability assessment components, the methods and the results, comments and suggestions. Those publications could contribute to this article’s discussion about useful research methods, definitions, framework and structures and data. Based on the comparison and information summary, several research gaps are figured out. So, we prefer to keep table 2 in the paper.      

Finally, thank you again for your valuable comments.

Kind regards,

Wenbin Ma

Reviewer 5 Report

The Authors have provided a valuable database search to summarize the current state of deep-sea mining sustainability research. They rightly point out several issues looming over this (so far speculative) industry, such as a lack of quantitative analyses or insufficient data on potential environmental impact of large-scale deep-sea operations.

"Since the concept of deep-sea mining was first proposed by John Mero (1970) in book The Mineral Resources of the Sea in 1965" (ca. line 75) - I don't understand, 1965 or 1970?

There is a number of minor language errors or odd turns of phrase. English grammar and spelling check is recommended, but overally the article is comprehensible.

My opinion for the Publisher is that the article can be published in the present form, although if the Authors desire so, it won't hurt if these two little issues are fixed.

Author Response

Dear Reviewer 5:

Thank very much for your comments. Please have a check of my responses and revision as follows.

Comment 1:The Authors have provided a valuable database search to summarize the current state of deep-sea mining sustainability research. They rightly point out several issues looming over this (so far speculative) industry, such as a lack of quantitative analyses or insufficient data on potential environmental impact of large-scale deep-sea operations.

Response: Thanks for your comments. Deep-sea mining, as an emerging coming industry, has attracted a lot of attention from academia and industry. Deep sea mining not only means a huge pie of economic benefits, but it may also bring challenges to the deep sea environment and even the global environment.  

Comment 2: "Since the concept of deep-sea mining was first proposed by John Mero (1970) in book The Mineral Resources of the Sea in 1965" (ca. line 75) - I don't understand, 1965 or 1970?

Response: Thanks for your comments. It is a bit confusion here. So, Page 1 Lines 28-31, the sentence is revised as ‘Since the concept of deep-sea mining was proposed in 1960s, its commercialization has still not been realized (Flipse, 1969; Mero, 1965)’.

Comment 3: There is a number of minor language errors or odd turns of phrase. English grammar and spelling check is recommended, but overall the article is comprehensible.

Response: Thanks for your comments. The language part of this article has undergone a further systematic revision and refinement.

Comment 4: My opinion for the Publisher is that the article can be published in the present form, although if the Authors desire so, it won't hurt if these two little issues are fixed.

Response: Thanks for your comments.

Finally, thank you again for your valuable comments.

Kind regards,

Wenbin Ma

Reviewer 6 Report

I have some serious difficulties with the set-up and the content of this manuscript. To summarize my main concerns:

1. There is major misalignment in title, objectives, content, literature review and conclusion. The structure of the manuscript is illogical (e.g. no clear results section, no clear discussion section and some confusing, almost similar sections (e.g. 2 and 4), in which literature is reviewed.

2. The grammar, writing style needs major revision, because in some key sections, the current text is non-informative, not concrete, or not to-the-point

3. The figures seem to have been taken quite randomly from the literature and - in my opinion - do not contribute to better understanding of the title/context of the manuscript.

4. No mentioning of the sustainable development goals has been made: a unique opportunity to compare details of terrestrial and deep-sea mining via life cycle assessments and learn from that. No concrete future directions have been made in the discussion (missing section) or conclusion

I provide a selection of examples from the manuscript text to support my findings.

The  abstract is vaguely formulated. The meaning and formulation of many sentences is unclear, e.g. “This article analyses the research content existing in the research topic of sustainable development of deep-sea mining from the perspective of the whole and the details.” (raises question) -This is one example, there are many more  throughout the manuscript: unclear formulations, no concrete information, unnecessary or illogical writing.

Another example from the introduction, page 1: “Since the concept of deep-sea mining was proposed in 1976, its commercialization has still not been realized (Mero, 1970)”. In a paper that pretty much leans on a literature review it seems odd that Mero in 1970 states something about a proposed issue in 1976.

On the first page: “The study reveals a key step in sustainability development of deep-sea mining is to obtain a series of comprehensive optimal indicators from a global perspective.” It is unclear what study in meant here, and the last part needs explanation, these so-called comprehensive optimal indicators from a global perspective raise many questions about their meaning.

 In lines 63-64 the objective is formulated: The objective of this paper is to analyze the issues existing in deep-sea mining sustainability development. It is unclear what issues are analyzed??? In addition, this objective seems different from what is written in the abstract and even in the conclusion another purpose is described. There seems to be no alignment.

The paper arrangement is confusing and not aligned with the topics described. For example, chapter 2 and 4 present almost the same topics, with many references, but it is unclear how all these literature ‘facts’ contribute to a clear objective or research questions or hypothesis.

Section 4  is introduced as a ‘discussion’ chapter in the introduction (lines 68/69: “Section 4 discusses the existing issues and challenges in deep-sea mining sustainability development”), but presents some methods  and results (table 2). Personally I feel that this kinds of long tabular information should go in a supplement and that the outcome should be condensed and presented in a shorter way. Besides: it is unclear how the authors reduced 90 articles, selected by keyword searches, to 30 articles that ….”could contribute to this articles’ discussion about useful research methods, definitions, framework and structures and data” (lines 261-263). No clear conclusion or discussions relate to this statement made related to the references in table 2.

In addition – it is not clear how the authors analyzed the table 2 information (lines 270 – 277) and I assume that the bulleted section starting at line 278 is a kind of result? To underpin the confusion, I address the mentioning of an “assessment index” or “deep-sea mining sustainability index”(lines 305 and 307). It is unclear what this index is, what the purpose is and in the conclusion (!!) another index “extreme environmental impact index” (line 405) is mentioned, which does not  make sense at all if it is not researched/analyzed/defined earlier.

The bulleted sentence for instance: “Parts of indicators are not independent, even contradictory. The couple research all these indicators together and get a comprehensive assessment index is missing” (lines 304-306) is confusing and indicators are hardly explained.

 Some figures raise some questions about their meaning/ purpose/context:

-      Figure 1 is mentioned in line 32, but not explained at all and the meaning is unclear. It is part of a deep sea mining life cycle addressed further in other figures, so no need to separately show this.

-      Figure 2 is taken from a master student on the internet, while better (e.g. ISA) sources are available (this figure is basically from Hein et al. 2013) – the three most important resources could be mentioned as well and their priorities in mining. I  noticed the lead author does show this figure in his master thesis with correct reference.

-      I am not sure what figure 3 contributes to this manuscript.

-      If figure 4 is used, the steps in this life cycle assessment should correspond with the impact assessment in figure 5 (they seem unrelated now) and somehow with the bulleted topics starting on page 13. If these topics are clearly outlined in the figures, and related to the findings in table 2, then a kind of coherent aligned text could be made. Now, the text, figures and tabular info seem loose descriptions of info that do not provide deeper insight in the topic.

The conclusion section is highly vague: 1. another purpose is mentioned, other than that in the introduction, 2. The main purpose seems here to discuss sustainable development: however, there is no section “Discussion” which is strange, 3. It does not concretely conclude on anything merely vague sentences and 4. At least a perspective for future development / direction of the topic should be presented, or discussed, but this is not present or highly disguised.

The reference list needs cleaning – I only mention no 52 and 56, of the lead author – which are the same.

Author Response

Dear Reviewer 6:

Thank very much for your comments. Please have a check of my responses and revision as follows.

Comment 1. There is major misalignment in title, objectives, content, literature review and conclusion. The structure of the manuscript is illogical (e.g. no clear results section, no clear discussion section and some confusing, almost similar sections (e.g. 2 and 4), in which literature is reviewed.

Response: Thanks for your comments. I am agreed with your opinion the structure of this article is a bit different with the traditional reviewing mode. This article is a literature review to analyze the status of sustainability development of deep-sea mining activities. This article analyses the research content existing in the research topic of sustainable development of deep-sea mining from the perspective of the whole and the details. Based on the results of the literature review, this paper also attempts to use the full life cycle assessment method to analyze the environmental impact of the entire process of deep-sea mining ore application. In the end, this paper summarizes the research gaps that still exist in the sustainable development of deep-sea mining. Section 1 is the introduction of the research background and significance of research. Section 2 gives a definition of deep-sea mining sustainability. Section 3 emphasizes on the current popular used method (life cycle assessment method) addressing deep-sea mining sustainability issue. This is also our next specific research direction. Also, in this section, we stated the importance of comparison between deep-sea mining with terrestrial mining. Section 4 gives a summary of collected literature to figure out the research gaps. As we all known, the topic of sustainable development of deep-sea mining itself is a very wide-ranging topic.       

Comment 2. The grammar, writing style needs major revision, because in some key sections, the current text is non-informative, not concrete, or not to-the-point.

Response: Thanks for your comments. The language part of this article has undergone a further systematic revision and refinement. Several correction examples are shown as follows (For more modification details, please see the original text):

On Page 1 Line 1-2, the title is changed as ‘Status of Sustainability Development of Deep-sea Mining Activities’.

On Page 1 Lines 14-15, the sentence is revised as ‘In the end, this paper summarizes the research gaps that exist in the sustainable development of deep-sea mining, including…’.

On Page 1 Lines 30-31, the sentence is revised as ‘The environmental pollution and sustainable development issues of deep-sea mining have become one of the biggest thresholds for its way forward’.

On Page 1 Lines 32-33, the sentence is revised as ‘Fig. 1 describes a schematic diagram of one typical deep-sea mining project’.

On Page 3 Lines 83-84, the sentence is revised as ‘Since the concept of deep-sea mining was first proposed, more than half a century later, it is still in the stage of exploration and experimental research’.

On Page 3 Lines 112-114, the following sentence is deleted ‘which would analyze its environmental and social impacts, economic profitability, production rate, working efficiency and social impacts’.

On Page 4 Lines 116-117, the sentence ‘find a compromise balance or an optimal balance between all influencing aspects’ is replaced by ‘find a compromise or an optimal balance among all influencing aspects’.

On Page 4 Line 136, ‘And’ is deleted.

On Page 5 Lines 158-161, the sentence is revised as ‘Regarding deep sea mining, the more urgent task is to scientifically analyze the environmental impact of its entire life cycle, the caused direct and indirect global and local problems, and also the research on environmental ecological restoration’.

On Page 5 Line 177, the phrase ‘both of them’ is added.

On Page 5 Line 179, the word ‘less’ is replaced by ‘shorter’.

On Page 6 Line 193, the punctuation is revised.

On Page 8 Line 225, the word ‘And’ is deleted.

On Page 9 Line 245, the word ‘topic’ is deleted.

On Page 14 Line 294, ‘land mining’ is replaced by ‘terrestrial mining’.

In Chapter 4 Lines 221-223, ‘Roche and Bice (2013) also analyzed the anticipated social and community impact on deep-sea mining development’ is revised as ‘Roche and Bice (2013) also analyzed the social and community assessment elements impact on deep-sea mining development’.

Comment 3: The figures seem to have been taken quite randomly from the literature and - in my opinion - do not contribute to better understanding of the title/context of the manuscript.

Response: Thanks for your comments. There are 6 figures in the paper. Figure 1 gives a schematic diagram of a typical deep-sea mining project. Figure 2 describes the deep-sea mining distribution around the world to show the great interest of governments, companies and research institutions in this field. Figure 3 is used to describe the vivid deep-ocean biological resources. I am agreed with your opinion. After thinking it over, figure 3 is deleted. Figure 4 describes the life cycle assessment method application in deep-sea mining. Figure 5 describes the environmental pollutions caused by deep-sea mining activities. Figure 6 is an explanation of deep-sea mining sustainability, which covers technological, economic, environmental and social aspects. After a round of inspection, I think these pictures can all help with the main idea, the research goal and the vivid portrayal of the deep-sea environment.    

Comment 4: No mentioning of the sustainable development goals has been made: a unique opportunity to compare details of terrestrial and deep-sea mining via life cycle assessments and learn from that. No concrete future directions have been made in the discussion (missing section) or conclusion.

Response: Thanks for your comments. I am agreed with your opinion. On Page 3 Lines 101-1069, the definition of sustainability is given as ‘a comprehensive concept connecting the ‘environmental sustainability’, ‘economic sustainability’, ‘biological sustainability’, ‘energy use sustainability’, which would analyze its environmental and social impacts, economic profitability, production rate, working efficiency and social impacts’. In this article, the environmental impact is focused. For instance, On Page 4 Lines 147-149, ‘Regarding deep sea mining, the more urgent task is to scientifically analyze the environmental impact of its entire life cycle, the direct and indirect global and local problems caused by the environmental damage, and the equally important research on environmental ecological restoration’. On the one hand, the research loopholes in the sustainable development of deep-sea mining found in the article are directions that we need to further explore in the future. Only when these aspects are solved, we can get a scientific and comprehensive evaluation parameter of sustainable development of deep-sea mining. In the section of conclusion Page 15-16 Lines 400-402, the following sentences for the future research is give: ‘In the next research, the author will apply the whole life cycle assessment method to quantify and compare the environmental pollution caused by deep sea mining and land mining to evaluate which mining mode is more environmentally friendly’.     

Comment 5: I provide a selection of examples from the manuscript text to support my findings. The abstract is vaguely formulated. The meaning and formulation of many sentences is unclear, e.g. “This article analyses the research content existing in the research topic of sustainable development of deep-sea mining from the perspective of the whole and the details.” (raises question).

Response: Thanks for your comments. I am totally agreed with your opinion. On Page 1 Lines 10-12, the sentence is revised as: ‘This article analyses the research of sustainable development of deep-sea mining from an overall perspective’.

Comment 6: Another example from the introduction, page 1: “Since the concept of deep-sea mining was proposed in 1976, its commercialization has still not been realized (Mero, 1970)”. In a paper that pretty much leans on a literature review it seems odd that Mero in 1970 states something about a proposed issue in 1976.

Response: Thanks for your comments. I am totally agreed with your opinion. On Page 1 Lines 28-30, the sentence is revised as: ‘Since the concept of deep-sea mining was proposed in 1960s, its commercialization has still not been realized (Flipse, 1969; Mero, 1965)’.

Comment 7: On the first page: “The study reveals a key step in sustainability development of deep-sea mining is to obtain a series of comprehensive optimal indicators from a global perspective.” It is unclear what study in meant here, and the last part needs explanation, these so-called comprehensive optimal indicators from a global perspective raise many questions about their meaning.

Response: Thanks for your comments. I am agreed with your opinion. On Page 1 Lines 43-44, the sentence is revised as: ‘The study reveals the importance of a set of comprehensive assessment indicators in the research of sustainability development of deep-sea mining’.

Comment 8: In lines 63-64 the objective is formulated: The objective of this paper is to analyze the issues existing in deep-sea mining sustainability development. It is unclear what issues are analyzed??? In addition, this objective seems different from what is written in the abstract and even in the conclusion another purpose is described. There seems to be no alignment.

Response: Thanks for your comments. I am agreed with your opinion. On Page 2 Lines 64-69, the sentence is revised as: ‘The objective of this paper is to analyze the research status of deep-sea mining (mainly focusing on polymetallic nodule mining activities) sustainability development. It attempts to use the full life cycle assessment method to analyze the environmental impact of the entire process of deep-sea mining ore application and summarizes the existing research gaps existing in this field’. On Page 17 Lines 408-411, the sentence is also revised accordingly.   

Comment 9: The paper arrangement is confusing and not aligned with the topics described. For example, chapter 2 and 4 presents almost the same topics, with many references, but it is unclear how all these literature ‘facts’ contribute to a clear objective or research questions or hypothesis. Section 4 is introduced as a ‘discussion’ chapter in the introduction (lines 68/69: “Section 4 discusses the existing issues and challenges in deep-sea mining sustainability development”), but presents some methods and results (table 2). Personally, I feel that this kind of long tabular information should go in a supplement and that the outcome should be condensed and presented in a shorter way.

Response: Thanks for your comments. Actually, in section 2, authors give the motivation and definition of deep-sea mining sustainability. And in section 4, through a comparison of existing literature, authors tried to figure out the current research gaps in this field. For table 2, authors will consult with Editor if it is suitable to set it as supplement.    

Comment 10: Besides: it is unclear how the authors reduced 90 articles, selected by keyword searches, to 30 articles that ….”could contribute to this articles’ discussion about useful research methods, definitions, framework and structures and data” (lines 261-263). No clear conclusion or discussions relate to this statement made related to the references in table 2.

In addition – it is not clear how the authors analyzed the table 2 information (lines 270 – 277) and I assume that the bulleted section starting at line 278 is a kind of result? To underpin the confusion, I address the mentioning of an “assessment index” or “deep-sea mining sustainability index”(lines 305 and 307). It is unclear what this index is, what the purpose is and in the conclusion (!!) another index “extreme environmental impact index” (line 405) is mentioned, which does not make sense at all if it is not researched/analyzed/defined earlier.

Response: Thanks for your comments. I am agreed with your opinion. On Page 10 Lines 262-264, the following sentence is revised as:’ Based on the keywords searching, roughly 90 publications are identified, and among those finally 30 publications are chosen (exclusion criteria including too old publication, not consistent with the research purpose, not formal publication on journal or conference, language not in English) those could contribute to this article’s discussion about useful research methods, definitions, framework and structures and data’. The collected publications in table 2 are compared based on criteria of research methods, research results, recommendations, and sustainability assessment indicators. On Page 14 Lines 308-310, the bulleted sentence is revised as: ‘Missing of relationship coupling research among these assessment indicator’. So, the assessment index is revised as ‘assessment indicator’. The deep-sea mining sustainability index is revised as: ‘deep-sea mining sustainability degree’ to avoid confusion. On Page 17 Lines 415, the ‘extreme environmental impact index’ is revised as: ‘maximum acceptable degree of environmental pollution’.      

Comment 11: The bulleted sentence for instance: “Parts of indicators are not independent, even contradictory. The couple research all these indicators together and get a comprehensive assessment index is missing” (lines 304-306) is confusing and indicators are hardly explained.

Response: Thanks for your comments. I agree with your opinion. This bulleted sentence is revised as: ‘Missing of relationship coupling research among these assessment indicators’. The ‘indicators’ in this sentence means the deep-sea mining sustainability assessment criteria.

Comment 12: Some figures raise some questions about their meaning/ purpose/context: Figure 1 is mentioned in line 32, but not explained at all and the meaning is unclear. It is part of a deep-sea mining life cycle addressed further in other figures, so no need to separately show this.

Response: Thanks for your comments. For figure 1, the following sentence is added on Page 1 Lines 33-34: ‘The structures involved in a deep-sea mining project consist of seabed mining vehicles, vertical lifting system, production support vessel, bulk carriers for the shipment, mineral ores processing and refining plant, etc., and Fig. 1 also describes the caused various environmental impact to the seabed, water columns and ocean surface’. We prefer to keep this image in the article as it provides a clear picture for readers who are not familiar with deep sea mining.

Comment 13: Figure 2 is taken from a master student on the internet, while better (e.g. ISA) sources are available (this figure is basically from Hein et al. 2013) – the three most important resources could be mentioned as well and their priorities in mining. I noticed the lead author does show this figure in his master thesis with correct reference.

Response: Thanks for your comments. The original reference is added and the title of the figure is revised as: ‘Fig. 2. Deep-sea mining target minerals distribution around the world (Hein et al., 2013)’. On Page 3 Lines 95, the following sentence is revised as ‘The typical minerals for deep-sea mining consists of polymetallic manganese nodules (most popularly focused), polymetallic sulphides, and cobalt-rich ferromanganese crusts, locating at different sea areas’.  

Comment 14: I am not sure what figure 3 contributes to this manuscript.

Response: Thanks for your comments. I am agreed with your opinion. After thinking it over, figure 3 is deleted.

Comment 15: If figure 4 is used, the steps in this life cycle assessment should correspond with the impact assessment in figure 5 (they seem unrelated now) and somehow with the bulleted topics starting on page 13. If these topics are clearly outlined in the figures, and related to the findings in table 2, then a kind of coherent aligned text could be made. Now, the text, figures and tabular info seem loose descriptions of info that do not provide deeper insight in the topic.

Response: Thanks for your comments. Figure 4 describes the selected life cycle assessment method, cradle-to-cradle mode, starting from the mineral ores mining to the end stage of waste treatment and material recycling. Figure 5 describes the related environmental impacts, which could be used the sustainability assessment indicators. They are exactly interconnected. Just as stated on Page 6 Lines 202-209, ‘Fig. 4 describes the life cycle impact assessment including the climate change, sediment plume, habitat physical destruction, toxic substances, benthic fauna and flora change, heavy metal concentration change, dissolved oxygen concentration change, sediment pore water change, noise and vibration influence, light pollution and tailings disposal, etc. These aspects could be used as the environmental indicators to present the pollution intensity by deep-sea mining’.   

Comment 16: The conclusion section is highly vague: 1. another purpose is mentioned, other than that in the introduction, 2. The main purpose seems here to discuss sustainable development: however, there is no section “Discussion” which is strange, 3. It does not concretely conclude on anything merely vague sentences and 4. At least a perspective for future development / direction of the topic should be presented, or discussed, but this is not present or highly disguised.

Response: Thanks for your comments. We agree with your opinion. The revisions are as follows. On Page 17 Lines 415-419, the following sentence is revised ‘The main purpose of this article is to discuss the research status of deep-sea mining sustainable development and summarize the existing research gaps, including the lack of environmental baseline data, environmental data detection systems and equipment, quantitative research on cumulative environmental impact, and lack of analysis and research on the maximum acceptable degree of environmental pollution’. Section 4 is the discussion and information summary section. In author opinion, the research gaps given in section 4 are just the guidelines for the future research. On Page 17 Lines, the following sentence for our team next stage research content is given ‘In the next research, the author will apply the whole life cycle assessment method to quantify and compare the environmental pollution caused by deep sea mining and land mining to evaluate which mining mode is more environmentally friendly’.   

Comment 17: The reference list needs cleaning – I only mention no 52 and 56, of the lead author – which are the same.

Response: Thanks for your comments. Duplicate reference has been removed.

Finally, thank you again for your valuable comments.

Kind regards,

Wenbin Ma

Reviewer 7 Report

see file below

Author Response

Dear Reviewer 7:

Thank very much for your comments. Please have a check of my responses and revision as follows.

Comment 1: The manuscript would be better called a review. The frequently used combination of words like “sustainability development analysis” or “sustainability development research” or “sustainable development” or The authors claim to have studied 90 relevant papers and focused on the most important 30 ones. The information of these papers is classified three categories (Table 2, four pages). The components of the publications, which are not clearly defined by the authors - the employed methods, and - results, comments, and suggestions.     

Response: Thanks for your comments. Firstly, this article belongs to a literature review type. This was selected at the beginning of our submission. Secondly, on Page 10 Lines 263-267, the literature selection and exclusion criteria are revised as follows ‘We consulted the ‘Web of Science’, ‘Google Scholar’ and ‘Scopus’ databases to review the literature related to deep-sea mining sustainability. All selected publications do not focus on a single aspect, but combine multiple aspects for analysis and research. The keywords that were used in our search were ‘deep-sea mining’, ‘sustainability development’, ‘sustainability assessment’, and ‘deep-sea resources exploitation’. Based on the keywords searching, roughly 90 publications are identified, and among those finally 30 publications are chosen (exclusion criteria including too old publication, not consistent with the research purpose, not formal publication on journal or conference, language not in English) those could contribute to this article’s discussion about useful research methods, definitions, framework and structures and data’. Thirdly, the components of the literature consist of 3 parts including the sustainability development assessment components, method and research results. On Page 13 Lines 274-276, the following sentence is added to explain the sustainability development components ‘Note: Deep-sea mining Sustainability Development Components: The information in this column indicates the range to which the sustainability assessment parameters belong to, such as environmental, ecological, technical, legal, economic aspects’.         

Comment 2: Does the content of that row is the conclusion of the original author of the published paper or is it added by the present authors? Eight sections follow, which describe different frequently overlapping aspects and mainly describe the common state of art: Exact data are missing, quantitative data are prevailing, more data are needed. Recycling is discussed, but the land-based refinery is not included. The sections overlap and their content confuses a reader. The theme of sustainability of the deep-sea mining is complex and not well defined. The presented compilation does not improve this situation. The announced priority setting of the problems is not achieved. The manuscript would also need careful English editing paying particular attention to grammar, spelling, and sentence structure so that the goals and results of the study are clear to the readers. And the consist of the contents and general rather unspecified remarks. In the next part some key processes of the sustainability

Response: Thanks for your comments. Firstly, the column of results, comments and suggestions in Table 2 is summarized from the selected publications. Secondly, the objective of this paper is to analyze the research status of deep-sea mining (mainly focusing on polymetallic nodule mining activities) sustainability development. It attempts to use the full life cycle assessment method to analyze the environmental impact of the entire process of deep-sea mining ore application and summarizes the existing research gaps existing in this field. Overall, it is a qualitative research. Thirdly, the research gaps figured out could be used a guideline for our future research. On Page 17 Lines 413-415, the following sentence is revised ‘In the next research, the author will apply the whole life cycle assessment method to quantify and compare the environmental pollution caused by deep sea mining and land mining to evaluate which mining mode is more environmentally friendly’. So, in our next stage research of applying life cycle assessment method in deep-sea mining activities, the relevant data analysis, quantitative calculation, and experimental simulation will be carried out. Fourthly, the definition of deep-sea mining sustainability development is given on Page 3 Lines 110-114 as follows ‘The sustainability applied in this thesis on DSM transport plans is a comprehensive concept connecting the ‘environmental sustainability’, ‘economic sustainability’, ‘biological sustainability’, ‘energy use sustainability’, which would analyze its environmental and social impacts, economic profitability, production rate, working efficiency and social impacts’.      

Comment 3: English language in this paper is corrected. Several correction examples are shown as follows (For more modification details, please see the original text):

On Page 1 Line 1-2, the title is changed as ‘Status of Sustainability Development of Deep-sea Mining Activities’.

On Page 1 Lines 14-15, the sentence is revised as ‘In the end, this paper summarizes the research gaps that exist in the sustainable development of deep-sea mining, including…’.

On Page 1 Lines 30-31, the sentence is revised as ‘The environmental pollution and sustainable development issues of deep-sea mining have become one of the biggest thresholds for its way forward’.

On Page 1 Lines 32-33, the sentence is revised as ‘Fig. 1 describes a schematic diagram of one typical deep-sea mining project’.

On Page 3 Lines 83-84, the sentence is revised as ‘Since the concept of deep-sea mining was first proposed, more than half a century later, it is still in the stage of exploration and experimental research’.

On Page 3 Lines 112-114, the following sentence is deleted ‘which would analyze its environmental and social impacts, economic profitability, production rate, working efficiency and social impacts’.

On Page 4 Lines 116-117, the sentence ‘find a compromise balance or an optimal balance between all influencing aspects’ is replaced by ‘find a compromise or an optimal balance among all influencing aspects’.

On Page 4 Line 136, ‘And’ is deleted.

On Page 5 Lines 158-161, the sentence is revised as ‘Regarding deep sea mining, the more urgent task is to scientifically analyze the environmental impact of its entire life cycle, the caused direct and indirect global and local problems, and also the research on environmental ecological restoration’.

On Page 5 Line 177, the phrase ‘both of them’ is added.

On Page 5 Line 179, the word ‘less’ is replaced by ‘shorter’.

On Page 6 Line 193, the punctuation is revised.

On Page 8 Line 225, the word ‘And’ is deleted.

On Page 9 Line 245, the word ‘topic’ is deleted.

On Page 14 Line 294, ‘land mining’ is replaced by ‘terrestrial mining’.

In Chapter 4 Lines 221-223, ‘Roche and Bice (2013) also analyzed the anticipated social and community impact on deep-sea mining development’ is revised as ‘Roche and Bice (2013) also analyzed the social and community assessment elements impact on deep-sea mining development’.

Finally, thank you again for your valuable comments.

Kind regards,

Wenbin Ma

Round 2

Reviewer 1 Report

It seems that the authors have considered the reviewers' comments. Therefore, this article can be published.

Reviewer 6 Report

not applicabel

Reviewer 7 Report

see below
